# Circulating Short-Chain Fatty Acids Are Positively Associated with Adiposity Measures in Chinese Adults

**DOI:** 10.3390/nu12072127

**Published:** 2020-07-17

**Authors:** Yiqing Wang, Huijun Wang, Annie Green Howard, Katie A. Meyer, Matthew C. B. Tsilimigras, Christy L. Avery, Wei Sha, Shan Sun, Jiguo Zhang, Chang Su, Zhihong Wang, Bing Zhang, Anthony A. Fodor, Penny Gordon-Larsen

**Affiliations:** 1Department of Nutrition, Gillings School of Global Public Health & School of Medicine, University of North Carolina at Chapel Hill (UNC-Chapel Hill), Chapel Hill, NC 27599, USA; yiqing@live.unc.edu (Y.W.); ktmeyer@email.unc.edu (K.A.M.); matthew_tsilimigras@unc.edu (M.C.B.T.); 2National Institute for Nutrition and Health, Chinese Center for Disease Control and Prevention, Beijing 102206, China; wanghj@ninh.chinacdc.cn (H.W.); zhangjg@ninh.chinacdc.cn (J.Z.); suchang@ninh.chinacdc.cn (C.S.); wangzh@ninh.chinacdc.cn (Z.W.); zhangbing@chinacdc.cn (B.Z.); 3Department of Biostatistics, Gillings School of Global Public Health, UNC-Chapel Hill, Chapel Hill, NC 27599, USA; aghoward@email.unc.edu; 4Carolina Population Center, UNC-Chapel Hill, Chapel Hill, NC 27516, USA; Christy_avery@unc.edu; 5Nutrition Research Institute, UNC-Chapel Hill, Kannapolis, NC 28081, USA; 6Department of Epidemiology, Gillings School of Global Public Health, UNC-Chapel Hill, Chapel Hill, NC 27599, USA; 7Department of Cancer Biostatistics, Levine Cancer Institute, Atrium Health, Charlotte, NC 28204, USA; wei.sha@atriumhealth.org; 8Department of Bioinformatics and Genomics, University of North Carolina at Charlotte (UNCC), Charlotte, NC 28223, USA; ssun5@uncc.edu (S.S.); afodor@uncc.edu (A.A.F.)

**Keywords:** short-chain fatty acids, BMI, waist-to-height ratio, fiber, gut metagenome

## Abstract

Epidemiological studies suggest a positive association between obesity and fecal short-chain fatty acids (SCFAs) produced by microbial fermentation of dietary carbohydrates, while animal models suggest increased energy harvest through colonic SCFA production in obesity. However, there is a lack of human population-based studies with dietary intake data, plasma SCFAs, gut microbial, and anthropometric data. In 490 Chinese adults aged 30–68 years, we examined the associations between key plasma SCFAs (butyrate/isobutyrate, isovalerate, and valerate measured by non-targeted plasma metabolomics) with body mass index (BMI) using multivariable-adjusted linear regression. We then assessed whether overweight (BMI ≥ 24 kg/m^2^) modified the association between dietary-precursors of SCFAs (insoluble fiber, total carbohydrates, and high-fiber foods) with plasma SCFAs. In a sub-sample (*n* = 209) with gut metagenome data, we examined the association between gut microbial SCFA-producers with BMI. We found positive associations between butyrate/isobutyrate and BMI (*p*-value < 0.05). The associations between insoluble fiber and butyrate/isobutyrate differed by overweight (*p*-value < 0.10). There was no statistical evidence for an association between microbial SCFA-producers and BMI. In sum, plasma SCFAs were positively associated with BMI and that the colonic fermentation of fiber may differ for adults with versus without overweight.

## 1. Introduction

Overall and central obesity are major risk factors for a wide range of chronic diseases, including cardiometabolic diseases [1,2,3]. As the prevalence of obesity has increased dramatically over the past decades around the world [1], many studies have been conducted to identify the biological determinants of obesity. Recent evidence has shown that the gut microbiota and microbiota-mediated metabolites like short-chain fatty acids (SCFAs) influence diet-induced obesity [4,5]. SCFAs like butyrate are major products of microbiota fermentation of dietary carbohydrates, especially soluble fiber and resistant starch [4]. In human studies, fiber-rich diets and Mediterranean diets have been shown to be positively associated with weight loss [6,7] and increased serum [8] and fecal SCFAs [9], respectively.

However, studies have yielded incongruent results for the SCFA-obesity association, which involves various factors like diet and gut microbiota. Whereas several studies have demonstrated that dietary SCFA supplementation may be beneficial to weight loss through appetite regulation [10,11] and increases in lipid oxidation and energy expenditure [12], others have suggested that SCFA production may promote obesity [5,13,14,15] through pathways including de novo lipogenesis [13] and energy harvesting from diet by gut microbiota [5]. For example, a randomized, controlled study showed that colonic delivery of SCFA propionate (i.e., oral supplementation of inulin-propionate ester) reduced energy intake, weight gain, and intra-abdominal fat accretion in overweight adults [10]. In contrast, in a study of obesity-prone mice fed a macronutrient-matched and isoenergetic high-fat diet, Isken et al. suggested that colonic SCFA production potentially outweighed the beneficial effects of soluble fiber supplementation on diet-induced obesity via contribution to increased digested energy [15]. Additionally, mouse model and in vitro assays suggest that the gut microbiome of mice and humans with (versus without) obesity had increased capacity to harvest energy through colonic fermentation of dietary carbohydrates and SCFA production [5,13].

There has been a lack of large population-based studies with large variation in dietary intake and gut microbiota composition to examine the incongruent experimental results in free-living people. Although a few community-based studies and case-control studies of fecal SCFAs in Western human populations support positive associations between SCFAs with overall body mass and central adiposity [16,17,18,19,20], few population-based studies have examined plasma SCFAs, which, in contrast to fecal SFCAs, may better represent the fraction of SCFAs that enter the host blood stream as a potential source of energy [21]. Therefore, we aimed to investigate the cross-sectional associations between plasma SCFAs with two adiposity measures, body mass index (BMI) and waist-to-height ratio (WHtR) in a socio-demographically diverse cohort of Chinese adults consuming a range of traditional and Western diets. We also assessed whether overweight and abdominal obesity modified the association between dietary precursors of SCFAs and plasma SCFAs. In a sub-sample with gut metagenome data, we examined the association between gut microbial SCFA producers with BMI and WHtR. 

## 2. Materials and Methods

### 2.1. Study Sample

We used data from the China Health and Nutrition Survey (CHNS), a population-based study across 12 provinces and three megacities that varied substantially in geography, economics, customs, social infrastructure, and health indicators, as previously described [22]. In 2015, we measured non-targeted plasma metabolomics from 500 participants aged 30–68 years from two adjacent southern provinces, Hunan and Guizhou. All 500 adults had anthropometry and diet data, and were thus eligible for the current study. We excluded participants if they were pregnant (*n* = 1) or had missing covariates (*n* = 9), resulting in an analysis sample of 490 adults, among which a subset of 209 adults also had gut metagenome data (Appendix A). The study met the standards for the ethical treatment of participants and was approved by the Institutional Review Boards of the University of North Carolina at Chapel Hill and the National Institute for Nutrition and Health, Chinese Center for Disease Control and Prevention. Informed consent was obtained for all participants. 

### 2.2. Plasma Short-Chain Fatty Acids (SCFAs)

Fasting blood samples were collected using venipuncture by trained clinicians with Ethylenediaminetetraacetic acid (EDTA) as an anticoagulant. The samples were immediately centrifuged to prepare plasma and stored at −80 °C until analyzed. All sites followed the same protocol for the collection, processing, and storage. The non-targeted metabolomics analysis was performed using Metabolon platform (Durham, NC, USA) consisting of a Waters ACQUITY ultrahigh-performance liquid chromatographer (LC, Milford, USA) coupled to a Thermo-Finnigan linear trap quadrupole (LTQ) mass spectrometer (MS, San Jose, CA, USA) at Metabolon’s partner campus in China [23]. This platform was designed and optimized around relative quantitation. Long et al. provides more detailed information on the Metabolon platform [24]. Briefly, plasma samples were extracted using methanol solvent and analyzed in concert with several types of quality controls: a pooled sample of each experimental sample served as a technical replicate; extracted water samples served as process blanks; and a cocktail of quality control standards that were carefully chosen not to interfere with the measurement of endogenous compounds were spiked into every analyzed sample, allowed instrument performance monitoring and aided chromatographic alignment. Instrument variability was determined by calculating the median relative standard deviation (RSD = 8%) for the internal standards that were added to each sample prior to injection into the mass spectrometers. Overall process variability was determined by calculating the median RSD (11%) for all endogenous metabolites (i.e., non-instrument standards) present in 100% of the pooled matrix samples. Experimental samples were randomized across the platform run with quality control samples spaced evenly among the injections. Signals in the metabolomics data were extracted and matched to the chromatographic data, mass-to-charge ratio, retention time/index in the Metabolon reference library of authenticated standards. We detected three SCFAs among the 1108 matched compounds: butyrate/isobutyrate, valerate, and isovalerate. Metabolon performed data normalizations by rescaling the area under the curve (AUC) of LC-MS peak for each SCFA within each run-day (i.e., batch run) to a median of one. No samples were below detection limits for butyrate/isobutyrate and isovalerate. The 136 samples below limit of detection for valerate were imputed with the minimum detected value (median-normalized AUC = 0.09). We log_2_ transformed the abundance of these three SCFAs. As the total SCFAs was also of interest, we summed up the raw AUC of all three SCFAs and log_2_ transformed the total SCFAs after it was rescaled to a median of one within each batch. 

### 2.3. Anthropometry

Anthropometry data were collected during physical examination by trained examiners. Weight was measured to the nearest 0.1 kg in light clothing using calibrated beam scales. Height was measured without shoes to the nearest 0.1 cm using portable stadiometers. Waist circumference was measured to the nearest 0.1 cm at midway between the lowest rib and iliac crest using non-elastic tape. We calculated BMI as weight divided by squared height (kg/m^2^) and WHtR as waist circumference divided by height. We defined overweight as BMI ≥ 24 kg/m^2^ and abdominal obesity as WHtR ≥ 0.5, according to the optimal cut-off points to indicate cardiovascular diseases risk in Chinese adults [25,26]. 

### 2.4. Diet

Individual dietary intake was recorded using three consecutive validated 24 h diet recalls by trained interviewers [27], including foods consumed at restaurants and other locations away from home. Household level food consumption was collected by calculating the changes in food inventory across the same three-day period and allocated to each member based on the proportions consumed. We estimated per capita daily intakes of nutrients based on a Chinese food composition table [27], in which insoluble fiber was measured by the neutral detergent method. We created an a priori high-fiber food group (Appendix A) consisting of: whole grains (e.g., millet), legumes (e.g., soybean curd), starchy roots (e.g., potato), vegetables (e.g., cabbage), mushrooms/seaweeds (e.g., Shitake mushroom), fruits (e.g., apple), and nuts/seeds (e.g., walnut). We grouped insoluble fiber, carbohydrate, and high-fiber foods by tertiles to limit the influence of extreme consumers, allow for non-linearity of relationships, and preserve statistical power. In an analysis of individual foods, we categorized those consumed by more than (i.e., legumes, starchy roots, vegetables) and less than 50% of the sample (i.e., whole grains, mushrooms/seaweeds, fruits, nuts/seeds) by median and any/no intake, respectively. 

### 2.5. Gut Metagenome

Stool samples were collected at home by participants who had been trained to use the QIAGEN collection kit (QIAGEN, Hilden, Germany). Samples were frozen at −20 °C immediately upon collection until processing. Stool DNA was extracted using TIANGEN DNA extraction kits (TIANGEN Biotech, Beijing, China), and sequenced on an Illumina platform with 150 paired-end (PE) by Novogene Bioinformatics Technology Co. Ltd., Tianjin, China in random order, so that batches were not related to collection centers. After filtering human DNA from the sequencing reads, we annotated the reads with MetaPhlAn2 based on the ChocoPhlAn pangenome database for taxonomic composition [28]. We selected 56 potential SCFA-producing microbiota species based on literature search (Appendix A) and calculated the total counts of the 56 selected species. We normalized and log_10_ transformed the raw counts of each species and the total counts of selected species using the following formula [29]: $\log_{10\;}\left(\frac{\mathrm{taxa}\;j\;\mathrm{count}\;\mathrm{for}\;\mathrm{sample}\;i}{\mathrm{total}\;\mathrm{taxa}\;\mathrm{count}\;\mathrm{in}\;\mathrm{sample}\;i}\;\times \;\mathrm{average}\;\mathrm{number}\;\mathrm{of}\;\mathrm{taxa}\;\mathrm{count}\;\mathrm{per}\;\mathrm{sample}+1\right)$. For analysis of specific species, we dichotomized 27 rare species that present in less than 25% of the sample to yes/no detected in the sample. 

### 2.6. Other Covariates

Sociodemographic and behavioral information were collected using interviewer-administered questionnaires. We estimated urbanization using a validated 12-component urbanization index that includes population density, health infrastructure, and transportation [30]. We calculated per capita household income by dividing household income by the number of household members. We measured total physical activity in metabolic equivalents (METs) per week using seven-day recalls of occupational, transportation, domestic, and leisure activities. We categorized urbanization index, per capita household income, and physical activity by tertiles. We dichotomized educational attainment by high school completion. We defined smokers as individuals who had ever smoked cigarettes and alcohol consumers as individuals who drank alcohol during past year. 

### 2.7. Statistical Analysis

We presented continuous variables as mean (standard deviation, SD) and categorical variables as number (proportion). We compared the characteristics by overweight and abdominal obesity using *t*-test for continuous variables and chi-square test for categorical variables. 

To determine the associations between plasma SCFAs with BMI and WHtR, we used a linear regression adjusting for the following covariates as guided by literature [4,31,32,33,34,35,36]: age, sex, batch run, province, urbanization, education, income, energy intake, insoluble fiber intake, physical activity, smoking, and alcohol intake. 

To investigate whether overweight and abdominal obesity modified the association between dietary precursors of SCFAs (insoluble fiber, total carbohydrates and high-fiber foods) with plasma SCFAs, we assessed the interaction of each of these dietary precursors with overweight and abdominal obesity in linear regression models of plasma SFCAs, using a Wald test at a nominal significance level of *p*-value < 0.10. In an exploratory analysis, we examined the interaction of each individual foods included in the high-fiber food group with overweight and abdominal obesity in linear regression models of plasma SCFAs. Then, in the sub-sample also containing gut metagenome data, we examined (1) the association between the overall microbial SCFA producers (i.e., SCFA-producing species) with BMI and WHtR using permutational multivariate analysis of variance (PERMANOVA) based on Bray–Curtis distance with 999 permutations [37] and (2) the association between the total relative abundance of all microbial SCFA producers with BMI and WHtR using linear regression. In an exploratory analysis, we examined each individual microbial SCFA producer using linear regression. For sensitivity, we tested the associations between microbial SCFA producers and plasma SCFAs. All analyses were adjusted for covariates described above. We corrected for multiple hypothesis testing for exploratory analyses of individual high-fiber foods and specific taxa using false discovery rate (FDR) [38]. We conducted sensitivity analysis by excluding participants who took antibiotics, pre/probiotics, or had diarrhea, irritable bowel syndrome (IBS), or inflammatory bowel disease (IBD), because these factors may affect the gut microbiome and therefore influence SCFA production and absorption. After applying these exclusion criteria, 462 adults remained in the analysis sample, among which 192 had gut metagenome data. We performed all statistical analyses in R 3.6.0 (R Core Team, Vienna, Austria). 

## 3. Results

The prevalence of overweight and abdominal obesity was 48.8% and 65.2%, respectively, in our analysis sample (Table 1). Adults with overweight were not different from those without overweight in terms of age, sex, plasma SCFAs abundance, province, urbanization, education, income, diet, physical activity, smoking, and alcohol intake. Adults with abdominal obesity were older, less physically active, and had higher abundance of plasma butyrate/isobutyrate, isovalerate, and total SCFAs than those without abdominal obesity.

We first examined the associations between plasma SCFAs and BMI and WHtR and found that butyrate/isobutyrate was positively associated with BMI (*p*-value = 0.04) and WHtR (*p*-value = 0.003), and isovalerate (*p*-value = 0.04) and total SCFAs (*p*-value = 0.03) were positively associated with WHtR (Table 2). For example, a fold increase of butyrate/isobutyrate was associated with a 0.40 and a 0.01 unit increase in BMI (kg/m^2^) and WHtR, respectively. 

Then, in models of plasma SFCAs, we tested the interaction between dietary precursors of SCFAs with overweight (Figure 1, Appendix A) and abdominal obesity (Figure 2, Appendix A). We observed effect modification of the association between insoluble fiber with butyrate/isobutyrate by overweight (interaction *p*-value = 0.095); between insoluble fiber with butyrate/isobutyrate (interaction *p*-value = 0.063) by abdominal obesity; and between carbohydrate with valerate (interaction *p*-value = 0.052) by abdominal obesity. Whereas the model estimated abundance of valerate was lower at high versus low carbohydrate consumption in people without abdominal obesity, valerate abundance was slightly higher at high versus low carbohydrate consumption in people with abdominal obesity. Moreover, when consuming middle level of insoluble fiber, adults with abdominal obesity had higher abundance of butyrate/isobutyrate and total SCFAs than those without abdominal obesity. 

In an exploratory analysis, we examined individual foods included in the high-fiber food group and observed effect modification (interaction *p*-value < 0.10) of the associations between legumes and fruits with valerate by overweight (Appendix A); between whole grains and fruits with valerate, and between nuts/seeds with butyrate/isobutyrate, isovalerate and total SCFAs by abdominal obesity (Appendix A). In general, consuming more of these fiber-rich foods tended to be associated with lower SCFAs in adults with abdominal obesity. After adjusting for FDR, only the interaction between fruits and abdominal obesity was nominally significant in the model of valerate (FDR-adjusted *p*-value = 0.07).

Finally, in the sub-sample with gut metagenome data, we tested the association between gut microbial SCFA producers with BMI and WHtR. We found little statistical evidence of association between the overall microbial SCFA producers with BMI (PERMANOVA R^2^ = 0.008, Table 3) or WHtR (PERMANOVA R^2^ = 0.005). There was no statistical evidence of association between the total relative abundance of microbial SCFA producers with BMI and WHtR either (*p*-value > 0.500). In an exploratory analysis examining the specific microbial SCFA producers, we found a few species that were associated with BMI and WHtR (Appendix A) at *p*-value < 0.05. For example, *Eubacterium hallii* and *Eubacterium rectale* were positively associated with BMI and WHtR. After adjusting for FDR, only the negative association between *Clostridium symbiosum* with BMI remained statistically significant (FDR adjusted *p*-value = 0.04). For sensitivity, we tested whether those species were associated with plasma SCFAs. We found an association between the overall microbial SCFA producers with total plasma SCFAs (PERMANOVA R^2^ = 0.01, *p*-value = 0.03, Appendix A), but not individual plasma SCFAs (R^2^ ranged 0.002–0.004, *p*-value > 0.05). We found no association between the total relative abundance (*p*-value > 0.05) of microbial SCFA producers with plasma SCFAs, although a few specific species were associated with butyrate/isobutyrate, valerate, isovalerate, and/or total plasma SCFAs (Appendix A) at *p*-value < 0.05, including *Faecalibacterium prausnitzii*. After adjusting for FDR, none of the association between specific microbiota species and plasma SCFAs was statistically significant (FDR adjusted *p*-value > 0.05).

In sensitivity analysis that restricted the sample to those who did not take antibiotics, pre/probiotics, or had diarrhea, IBS, or IBD, although the statistical significance reduced a little due to decreased sample size, the patterns of associations and parameter estimates remained similar for the associations between plasma SCFAs with BMI and WHtR (Appendix A); interactions of dietary precursors of SCFAs with overweight (Appendix A, Appendix A) and abdominal obesity in linear models of plasma SCFAs (Appendix A, Appendix A); and associations of the overall and total relative abundance of microbial SCFA producers with BMI and WHtR (Appendix A). 

## 4. Discussion

In our population-based cohort of Chinese adults, we found positive associations between plasma SCFAs and two adiposity measures, BMI and WHtR, independent of sociodemographic and behavioral factors, including urbanization, diet, and physical activity. Butyrate/isobutyrate was positively associated with BMI; and butyrate/isobutyrate, isovalerate, and total SCFAs were positively associated with WHtR. Our results provide insights into the potential role of SCFAs in the etiology of obesity and abdominal obesity. 

Several cross-sectional studies have reported positive associations between fecal SCFAs levels and obesity [16,17,18,19,20]. For example, a community-based study of 441 Colombian adults aged 18–62 years demonstrated that higher fecal butyrate, acetate, propionate and total SCFAs were associated with BMI, body fat, and waist circumference [17]. One hypothesis is that gut microbial dysbiosis in obesity may lead to less efficient SCFA absorption and, therefore, more SCFA excretion [17,39]. However, Rahat-Rozenbloom et al. [19] suggested that in 22 Canadian individuals aged >17 years, higher fecal acetate, butyrate, and total SCFAs in people with (versus without) overweight was not due to differences in diet or SCFAs absorption measured by the rectal dialysis bag method. Our findings of positive associations between plasma SCFAs and adiposity measures may support their results that higher SCFA excretion in higher body mass was not due to reduced SCFA absorption [19], although studies using both circulating and fecal SCFAs are needed to fully elucidate this hypothesis. Moreover, our findings show no difference in intakes of dietary precursors of SCFAs by overweight and abdominal obesity, but potential effect medication of associations between dietary precursors of SFCAs (e.g., insoluble fiber) and plasma SCFAs (e.g., butyrate/isobutyrate) by overweight and abdominal obesity, indicating that colonic fermentation of dietary precursors of SCFAs may differ for people with and without overweight, thereby leading to a different abundance of plasma SCFAs. Although we focused on well-established dietary precursors like fiber and carbohydrates, our results in butyrate/isobutyrate and isovalerate suggest that higher protein catabolism may be associated with higher adiposity, given that isobutyrate and isovalerate are major fermentation products of amino acids valine and leucine, respectively [40]. These branched-chain SCFAs function similarly to straight-chain SCFAs (e.g., butyrate) when modulating glucose and lipid metabolism [41].

Additionally, our results are consistent with Goffredo et al. [13], which showed that plasma concentrations of acetate, propionate, and butyrate were positively associated with body fat percentage and changes in BMI in 84 adolescents. Conversely, in 12 normal weight and overweight adults aged 18–65 years, Boets et al. [42] found that the rate of appearance of plasma propionate and butyrate measured by stable isotope dilution was lower in subjects with higher BMI. In 18 women who were obese, Layden et al. [43] found a negative association between serum acetate and visceral adipose tissue (although not BMI). These two studies had much smaller samples with less variation in BMI than our study. 

A potential reason for the positive relationship between SCFAs and obesity is that the gut microbiota of individuals with obesity may have higher capacity to harvest energy through SCFA production [5,13]. In fact, SCFAs are estimated to add about 10% of extra daily energy intake to adults eating a westernized diet [14]. Goffredo et al. [13] found that plasma SCFAs were associated with hepatic de novo lipogenesis and the gut microbiota of adolescents with obesity compared to their lean counterparts had higher capacity to ferment the same amount of fructose in vitro. Additionally, Yang et al. showed that the fecal microbiota of people with (versus without) obesity produced more propionate in response to in vitro fermentation of cereal grains [44]. Similarly, we found that carbohydrate consumption tended to be negatively associated with plasma valerate in adults without abdominal obesity, but tended to be positively associated with plasma valerate in adults with abdominal obesity, indicating that adults with higher abdominal adiposity may have higher potential for carbohydrate fermentation when consuming a carbohydrate-rich diet. The fact that fruits, whole grains, and nuts/seeds consumption tended to be negatively associated with SCFAs in adults with abdominal obesity in our study suggests that consuming these healthy fiber-rich foods may help decrease energy harvesting via SCFA production in adults with excess abdominal adiposity. Indeed, a double-blinded, randomized-controlled trial showed that fecal acetate, propionate and total SCFAs were lower in prebiotic dietary fiber-treated group than placebo group after treatment [45]. We also found different patterns of associations between different dietary precursors of SCFAs and plasma SCFAs. This may be because different gut microbiota compositions respond differently to different sources of fermentable fibers [44,46]. 

Nevertheless, not all epidemiological studies that support positive associations between SCFAs and adiposity measures also suggest the potential involvement of gut microbiota [13,16,17,19,20]. Whereas a recent meta-analysis of case-control studies found no evidence of association between obesity and phyla richness [16], other studies found that higher fecal SCFAs were associated with lower microbiota diversity, as well as higher gut permeability, Firmicutes/Bacteriodetes ratio and cardiometabolic disease-associated taxa [17,19]. While a study of 96 adolescent girls found no association between microbiota phyla abundance and fecal SCFAs [20], a study 84 adolescent boys and girls found a positive association between plasma SCFAs and obesity-related microbiota, including *Faecalibacterium*, *Streptococcus* and *Actinomyces* [13]. Although our sub-sample analysis using gut metagenome data showed that a few specific microbial SCFA producers were associated with BMI, WHtR, and plasma SCFAs, only the negative association between *Clostridium symbiosum* and BMI reached statistical significance after considering multiple testing (FDR adjusted *p*-value = 0.04). The relationships between gut microbiota with SCFAs production and host adiposity are complicated given the convoluted metabolic pathways and bacterial cross-feeding interactions [47]. It is possible that we lacked statistical power to detect these associations, given that our sample comprised free-living people from a range of rural and urban communities, providing more diversity in environments and behaviors, than previous studies that were conducted in a single city or neighborhood [17,19]. 

The strengths of our study include the well-characterized, population-based cohort, high-quality dietary data from three-consecutive 24-h recalls and household food inventories, and rich host factors collected from standardized protocols, allowing us to control for a wide range of potential confounders. Additionally, to our knowledge, we are the first study to examine the associations between plasma SCFAs, gut metagenome, and adiposity measures in population-based adult cohort and our study is relatively larger than previous epidemiological studies of SCFAs [16,17,18,19,20]. However, it is possible that we lacked statistical power to test potential interactions between dietary precursors of SCFAs with overweight and abdominal obesity in models of plasma SCFAs. After considering multiple testing, none of the interactions reached statistical significance (FDR adjusted *p*-value > 0.05). Other limitations of our study include the (1) cross-sectional design; (2) potential measurement error in diet and lack of detailed information on fiber types to distinguish fermentable versus non-fermentable fiber; (3) selected microbial SCFA producers may not comprehensively capture the SCFA-producing ability of the gut microbiota; (4) lack of internal standards for SCFAs to determine the concentrations for SCFAs and could not detect two major SCFAs (acetate and propionate) due to small molecular sizes. The fraction of acetate:propionate:butyrate in the portal system is approximately 69:23:8 [48]. Since butyrate could be undetectable in circulation because of rapid usage [49], we could not exclude the possibility that we underestimated the abundance of plasma butyrate and other SCFAs. Lastly, our results in dietary precursors of SCFAs may have limited generalizability with regard to other populations with different dietary habits and dietary sources of fiber.

## 5. Conclusions

Our study in a population-based cohort of Chinese adults suggests that plasma SCFAs may be positively associated with BMI and WHtR, providing insights into the possible role of SCFAs in obesity etiology. Our findings suggest differential associations between dietary precursors of SCFAs and plasma SFCAs for adults with (versus without) overweight and abdominal obesity, indicating that colonic fermentation of carbohydrates and SCFA production as a source of dietary energy may be differ for adults with versus without overweight and abdominal obesity. However, we found little statistical evidence for an association between the relative abundance of microbial SCFA producers with BMI and WHtR. Further studies with longitudinal data and comprehensive measurement of SCFAs are needed to replicate our results for relationship between plasma SCFAs and adiposity. Well-designed randomized-controlled trails are also needed to clarity whether gut microbiota provide differential energy harvests in people with different levels of body weight and adiposity.

## Figures and Tables

**Figure 1 nutrients-12-02127-f001:**
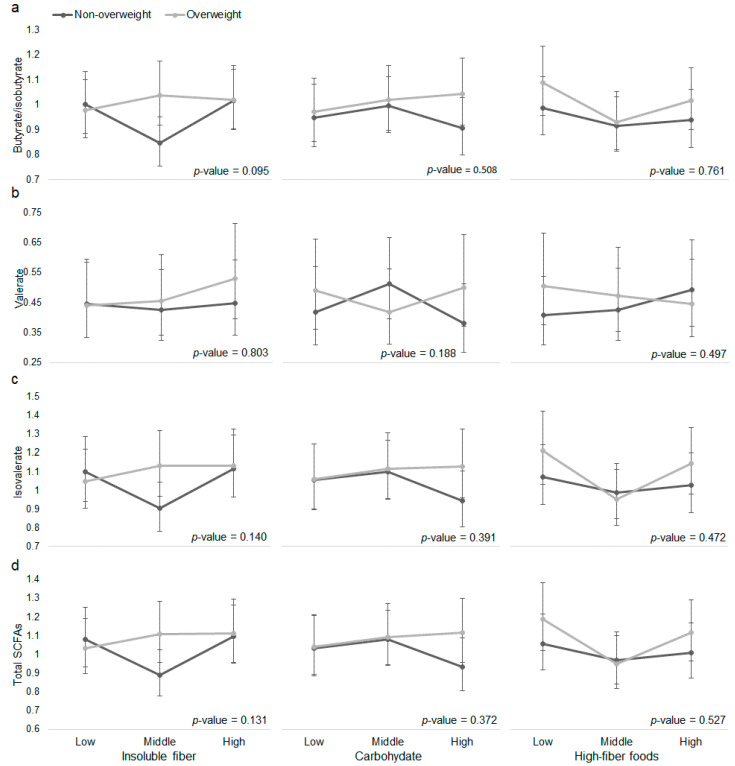
The associations between dietary precursors of short-chain fatty acids (SCFAs) and plasma (**a**) butyrate/isobutyrate, (**b**) valerate, (**c**) isovalerate, and (**d**) total SCFAs by overweight. Overweight: BMI ≥ 24 kg/m^2^. Vertical axes represent model predicted (marginal means) plasma SCFAs abundance. Dietary intakes of insoluble fiber, total carbohydrates, and high-fiber foods were categorized by tertiles to represent low, middle, and high intakes. Linear model was adjusted for age, sex, batch run, province, urbanization, income, education, physical activity, total energy intake, alcohol, and ever smoking. For analysis of total carbohydrates and high-fiber foods, insoluble fiber intake was additionally adjusted in model. *p*-value for the interaction between each dietary precursor of SCFAs and overweight was derived using a Wald test. *p*-value > 0.05 for all comparisons of plasma SCFA abundance at a given level of a dietary precursor by overweight.

**Figure 2 nutrients-12-02127-f002:**
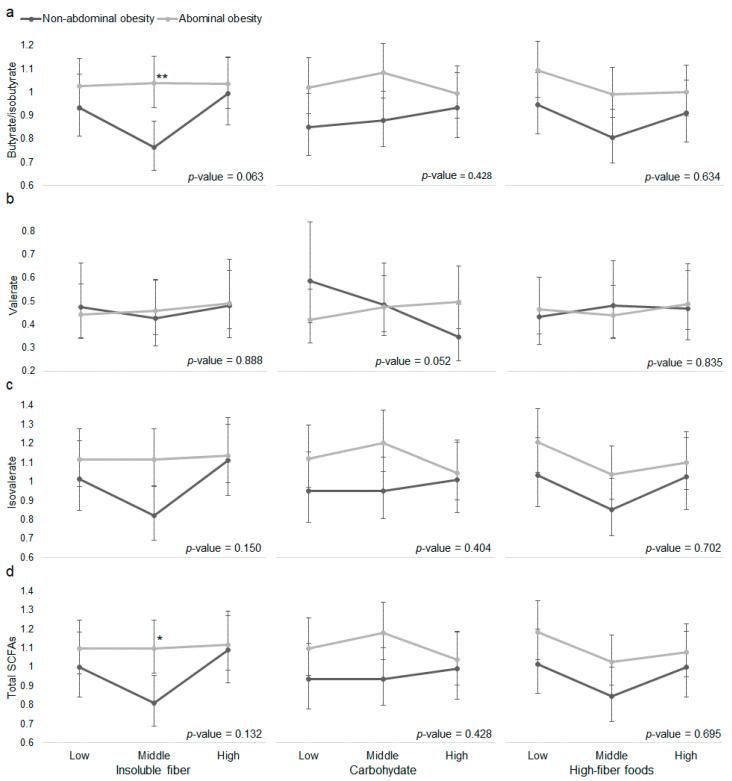
The associations between dietary precursors of short-chain fatty acids (SCFAs) and plasma (**a**) butyrate/isobutyrate, (**b**) valerate, (**c**) isovalerate, and (**d**) total SCFAs by abdominal obesity. Abdominal obesity: waist-to-height ratio ≥ 0.5. Vertical axes represent model predicted (marginal means) SCFAs abundance. Dietary intakes of insoluble fiber, total and carbohydrate, and high-fiber foods were categorized by tertiles to represent low, middle, and high intakes. Linear model was adjusted for age, sex, batch run, province, urbanization, income, education, physical activity, total energy intake, alcohol, and ever smoking. For analysis of total carbohydrate and high-fiber foods, insoluble fiber intake was additionally adjusted in model. *p*-value for the interaction between each dietary precursor of SCFAs and abdominal obesity was derived using a Wald test. *, *p*-value < 0.5; **, *p*-value < 0.01 for comparisons of plasma SCFAs abundance at a given level of dietary precursor by abdominal obesity.

**Table 1 nutrients-12-02127-t001:** Characteristics of the metabolomics analysis sample by overweight and abdominal obesity.

	Overweight ^1^		Abdominal Obesity ^1^	
	Without	With	Without	With
*n* (%)	251 (51.2%)	239 (48.8%)	170 (34.8%)	318 (65.2%)
Age, years	52.3 (9.0)	52.2 (9.1)	51.0 (8.8)	52.9 (9.1) *
Women, *n* (%)	155 (61.8%)	135 (56.5%)	100 (58.8%)	189 (59.4%)
Body mass index (BMI), kg/m^2^	21.8 (1.7)	26.6 (2.0) ***	21.1 (2.0)	25.5 (2.7) ***
Waist-to-height-ratio (WHtR)	0.5 (0.05)	0.6 (0.05) ***	0.5 (0.03)	0.6 (0.04) ***
Butyrate/isobutyrate ^2^	−0.05 (0.7)	−0.001 (0.8)	−0.2 (0.7)	0.04 (0.7) **
Valerate ^2^	−1.1 (1.8)	−1.1 (1.7)	−1.1 (1.7)	−1.0 (1.8)
Isovalerate ^2^	0.07 (0.9)	0.1 (1.0)	−0.03 (0.9)	0.2 (0.9) *
Total short-chain fatty acids (SCFAs) ^2^	0.04 (0.9)	0.09 (0.9)	−0.06 (0.9)	0.1 (0.9) *
Hunan province	159 (63.3%)	145 (60.7%)	102 (60.0%)	200 (62.9%)
Urbanization index, *n* (%) ^3^				
(39.2–64.2)	89 (35.5%)	84 (35.1%)	63 (37.1%)	110 (34.3%)
Middle (64.2–81.5)	77 (30.7%)	84 (35.1%)	51 (30.0%)	109 (34.3%)
High (81.5–99.6)	85 (33.9%)	71 (29.7%)	56 (32.9%)	99 (31.1%)
Completed high school education	85 (33.9%)	65 (27.2%)	52 (30.6%)	96 (30.2%)
Per capita household income, *n* (%) ^4^				
Low (0–10 k yuan)	89 (35.5%)	74 (31.0%)	60 (35.3%)	102 (32.1%)
Middle (10–22.1 k yuan)	85 (33.9%)	79 (33.1%)	65 (38.2%)	99 (31.1%)
High (22.1–468 k yuan)	77 (30.7%)	86 (36.0%)	45 (26.5%)	117 (36.8%)
Total energy, 1000 kcal ^5^	1.9 (0.6)	1.9 (0.7)	1.9 (0.6)	1.9 (0.7)
Insoluble fiber intake, *n* (%) ^5^				
Low (1.5–8.2 g)	77 (30.7%)	87 (36.4%)	58 (34.1%)	105 (33.0%)
Middle (8.2–12.5 g)	88 (35.1%)	75 (31.4%)	57 (33.5%)	105 (33.0%)
High (12.5–69.7 g)	86 (34.3%)	77 (32.2%)	55 (32.4%)	108 (34.0%)
Carbohydrate intake, *n* (%) ^5^				
Low (65.2–172 g)	83 (33.1%)	81 (33.9%)	55 (32.4%)	108 (34.0%)
Middle (172–248 g)	87 (34.7%)	76 (31.8%)	61 (35.9%)	101 (31.8%)
High (248–649 g)	81 (32.3%)	82 (34.3%)	54 (31.8%)	109 (34.3%)
High-fiber foods, *n* (%) ^5^				
Low (0–344 g)	87 (34.7%)	77 (32.2%)	61 (35.9%)	102 (32.1%)
Middle (344–482 g)	85 (33.9%)	78 (32.6%)	55 (32.4%)	108 (34.0%)
High (482–1200 g)	79 (31.5%)	84 (35.1%)	54 (31.8%)	108 (34.0%)
Physical activity, *n* (%) ^6^				*
Low (0–50 METS/wk),	76 (30.3%)	84 (35.1%)	43 (25.3%)	117 (36.8%)
Middle (50–147 METS/wk)	82 (32.7%)	83 (34.7%)	58 (34.1%)	106 (33.3%)
High (147–1390 METS/wk)	93 (37.1%)	72 (30.1%)	69 (40.6%)	95 (29.9%)
Ever smoking, *n* (%)	100 (39.8%)	93 (38.9%)	74 (43.5%)	118 (37.1%)
Drank alcohol last year, *n* (%)	62 (24.7%)	64 (26.8%)	38 (22.4%)	87 (27.4%)

Continuous variables [mean (SD)] were tested by *t*-test and categorical variables [*n* (%)] were tested by chi-square test. *, *p*-value < 0.05; **, *p*-value < 0.01; ***, *p*-value < 0.001 when comparing participants with versus without overweight and abdominal obesity. ^1^ Overweight: BMI ≥ 24 kg/m^2^; abdominal obesity: waist-to-height ratio ≥0.5. ^2^ Plasma SCFAs were measured by non-targeted metabolomics, which only provided relative quantitation, rather than absolute concentrations of SCFAs. The abundance for each SCFA was scaled to a median of one and log_2_ transformed. ^3^ Urbanization index encompasses 12 dimensions of urbanization, including population density, health infrastructure, and transportation. Urbanization index was categorized by tertiles to represent low, middle, and high levels of urbanization. ^4^ Per capita household income was estimated by dividing the household income by the number of household members. Per capita household income was categorized by tertiles to represent low, middle, and high levels of income. ^5^ Dietary intakes were measured by 3-consecutive 24 h dietary recalls and household food inventories. The intake of high-fiber foods was calculated as the sum of whole grains, legumes, starchy roots, vegetables, mushrooms/seaweeds, fruits, nuts/seeds. Insoluble fiber, carbohydrate, and high-fiber food score were categorized by tertiles to represent low, middle, and high intakes. ^6^ Physical activity was estimated by 7-day physical activity recalls in METS and was categorized by tertiles to represent low, middle, and high levels of physical activity.

**Table 2 nutrients-12-02127-t002:** The associations between plasma short-chain fatty acids (SCFAs) with body mass index (BMI) and waist-to-height ratio (WHtR).

		BMI (*n* = 490)		WHtR (*n* = 488)	
	Mean (SD)	β (95% Confidence Interval)	*p*-Value	β (95% Confidence Interval)	*p*-Value
Butyrate/isobutyrate	−0.03 (0.75)	0.40 (0.01, 0.78)	0.04	0.01 (4 × 10^−3^, 0.02)	0.003
Valerate	−1.09 (1.75)	−0.01 (−0.17, 0.16)	0.93	1 × 10^−3^ (−2 × 10^−3^, 4 × 10^−3^)	0.48
Isovalerate	0.09 (0.94)	0.20 (−0.10, 0.52)	0.18	0.01 (3 × 10^−4^, 0.01)	0.04
Total SCFAs	0.07 (0.89)	0.24 (−0.09, 0.56)	0.15	0.01 (8 × 10^−4^, 0.01)	0.03

The mean (standard deviation, SD) for BMI (kg/m^2^) and WHtR was 24.01 (3.18) and 0.52 (0.06), respectively. Because the plasma SCFA abundance was log_2_ transformed, the linear model coefficients are interpreted as units of BMI and WHtR associated with a fold increase in a SCFA. Model was adjusted for age, sex, batch run, province, urbanization, income, education, physical activity, total energy intake, insoluble fiber intake, alcohol, and ever smoking.

**Table 3 nutrients-12-02127-t003:** The associations between the overall and the total relative abundance of 56 microbial short-chain fatty acid (SCFA) producers with body mass index (BMI) and waist-to-height-ratio (WHtR).

			Overall ^1^		Total ^2^	
	*n*	Mean (SD)	R^2^	*p*-Value	β (95% CI)	*p*-Value
BMI	209	24.35 (3.21)	0.008	0.05	−0.04 (−1.7, 1.61)	0.96
WHtR	208	0.53 (0.06)	0.005	0.30	0.00 (−0.03, 0.03)	0.78

The 56 microbial SCFA producers were selected from literature and the full list with references is in Appendix A. The raw counts of each species and the total counts of the 56 species were normalized and log_10_ transformed [29]. Model was adjusted for age, sex, province, urbanization, income, education, total energy, insoluble fiber intake, physical activity, smoking, and alcohol intake. ^1^ R^2^ and *p*-value were calculated using permutational multivariate analysis of variance (PERMANOVA) of all 56 species. ^2^ Linear regression was performed on the total relative abundance of the 56 species.

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
