# Peer review of "Circulating Short-Chain Fatty Acids Are Positively Associated with Adiposity Measures in Chinese Adults"

_nutrients, 2020, doi:10.3390/nu12072127_

Round 1

Reviewer 1 Report

Wang and colleagues have provided a manuscript entitled “Circulating Short-chain Fatty Acids are Positively Associated with Adiposity Measures in Chinese Adults”.

In a cohort of 490 Chinese adults, they investigated the associations of plasma short chain fatty acids with BMI, measures of central obesity and nutritional intake questionnaire data. In a subset of the samples, gut metagenome data was also analysed.

Plasma SCFA were measured using a common metabolomics platform provider, Metabolon. In section 2.2, it is not quite clear what was meant is the sentence: “Metabolon rescaled the raw area count of each metabolite within each run-day to a median of one to correct for differences in instrument inter-day tuning…”. It is commonly acknowledged that mass spectrometry methods can suffer inter-day variability, however internal standards are generally used to correct for this. As the authors state the “metabolomics could not provide concentrations”, I assume no internal standards available/used for SCFA measurements? There are numerous other batch correction procedures that would likely help generate more accurate data. In addition, response factors could help improve the quantitative nature of the measurements.

Further, if each metabolite was rescaled to a median of one, the sum of these is no longer an accurate reflection of the total plasma SCFA.

It’s not indicated what units the SCFA are in for Table 1. If these are concentrations (x moles or grams per Litre), you can avoid negatives after log2 transformation by changing the prefix e.g. 0.1 mM = 100 uM. Have these been rescaled to a median of one? Butyrate, isovalerate and the total appear to have been, although if valerate has, it would appear to have a highly skewed distribution.

Where appropriate quality control measures used in the analysis? Where the samples randomized prior to analysis? What was the coefficient of variation for quality control samples?

Similar to the above concerns, it’s common to impute metabolites below the limit of detection. However, it should be stated how many samples this was performed in and what the limit of detection is compared to the sample concentrations.

The statistical analysis performed is appropriate. The authors have adjusted for confounders in their linear regressions between SCFA and BIM/WHR. To investigate whether being overweight modified the association, they introduced an interaction term, assessing it at an increased significance threshold. For the metagenome data, a non-parametric multivariate ANOVA was used in combination with univariate analysis with linear regression.

Unfortunately, the authors only briefly consider multiple testing correction. Having said that, there would likely be large correlations between the 4 SCFA and 2 adiposity measures, suggesting Bonferroni correction may be overly strict.

In several places, p-values are indicated at p<0.05 in text. It is preferable to indicate the actual p-value of analyses where possible, as has been done in Table 2 (although it would be preferable to convert all p-values to scientific notation).

Overall, the authors have produced a well written manuscript, with detailed introduction and discussion sections.

Reviewer 2 Report

I read the manuscript with interest, entitled "Circulating Short-chain Fatty Acids in Chinese Adults Positively Associated with Adiposity Measures," by Wang at.al.

In this paper, the authors examined a relatively large-population study on the associations between plasma short-chain fatty acids (SCFAs) in a socio-demographically diverse cohort of Chinese adults consuming traditional and Western diets with two adiposity measurements, body mass index (BMI) and waist-to-height ratio (WHtR). In addition, they have studied the association between gut microbial SCFA producers with BMI and WHtR. The results are interesting and will give insights into the possible role of SCFAs in the etiology of obesity. The manuscript is well written, with good data presentation, Supporting Information, and cited all the relevant references.

I have just a couple of minor corrections below.

  • “Figure.1” is repeated twice – On page 8, the figure should be “Figure.2”.
  • On page 9 – line 281, Figure S2 should be Figure S3.

Author Response

Response to Reviewer 2 Comments

We appreciate the comments of the reviewer.

Point 1: “Figure.1” is repeated twice – On page 8, the figure should be “Figure.2”.

Response 1: Corrected.

Point 2: On page 9 – line 281, Figure S2 should be Figure S3.

Response 2: Corrected.